Involvement of plasminogen activator inhibitor-1 in p300/p53-mediated age-related atrial fibrosis

Lai Yingyu 1 2 3
He Jintao 4
Gao Xiaoyan 1 2
Peng Dewei 1 2
Zhou Huishan 1 2
Xu Yuwen 1 2
Luo Xueshan 1 2
Yang Hui 1 2
Zhang Mengzhen 1 2
Deng Chunyu 1 2
Wu Shulin 1 2
Xue Yumei 1 2
Zhou Feng stacy85@163.com 5
Rao Fang raofang@gdph.org.cn 1 2 4
1 Medical Research Institute, Guangdong Provincial Key Laboratory of Clinical Pharmacology, Guangdong Provincial People’s Hospital (Guangdong Academy of Medical Sciences), Southern Medical University , Guangzhou , Guangdong , China
2 Guangdong Cardiovascular Institute, Guangdong Provincial People’s Hospital, Guangdong Academy of Medical Sciences , Guangzhou , Guangdong , China
3 Department of Pharmacy, The People’s Hospital of Hezhou , Hezhou , Guangxi , China
4 School of Medicine, South China University of Technology , Guangzhou , Guangdong , China
5 Department of Neurology, The Fifth Affiliated Hospital of Sun Yat-sen University , Zhuhai , Guangdong , China
Korla Praveen
Electronic publication date: 2023 Dec 12
Publication date: 2023
Volume: 11
Electronic Location ID: e16545
Received 2023 Aug 29; Accepted 2023 Nov 8
Copyright: ©2023 Lai et al.
Copyright year: 2023
Copyright holder: Lai et al.
License: This is an open access article distributed under the terms of the Creative Commons Attribution License, which permits unrestricted use, distribution, reproduction and adaptation in any medium and for any purpose provided that it is properly attributed. For attribution, the original author(s), title, publication source (PeerJ) and either DOI or URL of the article must be cited.
License URL: https://creativecommons.org/licenses/by/4.0/

Keywords: Plasminogen activator inhibitor-1, p300, p53, Bioinformatics analysis, Age-related atrial fibrosis, Atrial fibrillation, Thrombosis, Fibrinolytic system, Senescence

Funding: National Natural Science Foundation of China 82270321 Guangdong special funds for Science and Technology Innovation Strategy, China Guangdong Province-GDCI 2021 Science and Technology Program of Guangdong Province 2019B020230004 This work was supported by the National Natural Science Foundation of China (No. 82270321), the Guangdong special funds for Science and Technology Innovation Strategy, China (Stability support for scientific research institutions affiliated to Guangdong Province-GDCI 2021) and the Science and Technology Program of Guangdong Province (No. 2019B020230004). The funders had no role in study design, data collection and analysis, decision to publish, or preparation of the manuscript.

==============================
Plasminogen activator inhibitor-1 (PAI-1), a key regulator of the fibrinolytic system, is also intimately involved in the fibrosis. Although PAI-1 may be involved in the occurrence of atrial fibrillation (AF) and thrombosis in the elderly, but whether it participated in aging-related atrial fibrosis and the detailed mechanism is still unclear. We compared the transcriptomics data of young (passage 4) versus senescent (passage 14) human atrial fibroblasts and found that PAI-1 was closely related to aging-related fibrosis. Aged mice and senescent human and mouse atrial fibroblasts underwent electrophysiological and biochemical studies. We found that p300, p53, and PAI-1 protein expressions were increased in the atrial tissue of aged mice and senescent human and mouse atrial fibroblasts. Curcumin or C646 (p300 inhibitor), or p300 knockdown inhibited the expression of PAI-1 contributing to reduced atrial fibroblasts senescence, atrial fibrosis, and the AF inducibility. Furthermore, p53 knockdown decreased the protein expression of PAI-1 and p21 in senescent human and mouse atrial fibroblasts. Our results suggest that p300/p53/PAI-1 signaling pathway participates in the mechanism of atrial fibrosis induced by aging, which provides new sights into the treatment of elderly AF.

Introduction

Atrial fibrillation (AF) is most common arrhythmia in the clinic, and is strongly associated with increased morbidity and mortality of cardiovascular diseases. The incidence of AF increases gradually with age. It imposes a huge public health burden on aging society (Kornej et al., 2020). The characteristic of AF is progressive in nature. However, the current treatment options are mainly symptomatic. So, the prevention of AF progression is only moderately effective, especially in the elderly. Thus, there is an urgent need for developing upstream therapies directed at root causes of AF (Calvo, Filgueiras-Rama & Jalife, 2018; Li & Brundel, 2020). As a transcriptional coactivator, the role of p300 in aging and cardiovascular diseases has drawn increasing attention (Ghosh, 2020; Rai et al., 2019). Previously, we were concerned that aggravated aging-related atrial fibrosis is a leading cause of increased susceptibility to AF in elderly (Henry et al., 2016; Ribeiro Mesquita et al., 2020). And our previous study has demonstrated that p300/p53 pathway is the potential mechanism involved in atrial fibroblast senescence and aging-related atrial fibrosis. Inhibition of p300 can significantly rescue atrial fibroblast senescence and aging-related atrial fibrosis (Gao et al., 2023). Acetyltransferase p300 is a new target for the prevention and treatment of AF. But the relevant molecular regulatory mechanisms are far from being elucidated.

In this study, replicative senescence model of human atrial fibroblasts was established through cell passage, mRNA-sequencing (mRNA-seq) analysis was performed to compare the transcriptomic changes between P4 (young) and P14 (senescent) human atrial fibroblasts. We found that plasminogen-activator inhibitor 1 (PAI-1) was significantly increased in senescent human atrial fibroblasts. As is well known, PAI-1 is the principal inhibitor of urokinase plasminogen activator (uPA) and tissue plasminogen activator (tPA) in the fibrinolysis system (Van De Craen, Declerck & Gils, 2012). Elevated levels of PAI-1 lead to decreased fibrinolytic activity and the hypercoagulable state of blood, which increase the risk of thromboembolism disease (Dawson & Henney, 1992). Research has indicated that elevated circulating PAI-1 and thrombin-antithrombin (TAT) were significantly associated with increased risk of stroke in patients with AF (Wu et al., 2015). In addition, studies have found that PAI-1 is also a marker and mediator of cell senescence, and plays a pivotal role in the fibrosis (Baumeier et al., 2021; Rana et al., 2020). AF, especially in elderly patients, manifests as both atrial fibrosis and hypercoagulability, suggesting that PAI may play an important role in promoting fibrosis and leading to hypercoagulability. Although previous studies have confirmed that PAI-1 has multifunctional functions, including coagulation promoting, fibrosis, and aging, its regulatory mechanism in aging-related atrial fibrosis is still unclear.

PAI-1 has been shown to be a critical downstream target of p53 in the induction of cellular senescence and fibrosis (Kortlever, Higgins & Bernards, 2006; Eren et al., 2014; Li et al., 2021). As is well known, p53 is an important molecule in the aging signaling pathway, which can cause cell senescence by activating downstream p21 (Chen et al., 2020). During the aging process, on one hand, p53 may promote the senescence of atrial fibroblasts by activating p21, and on the other hand, promote the formation and deposition of extracellular matrix by activating PAI-1, as well as exacerbate fibrinolytic dysfunction. Therefore, in the present study, we speculate that p300/p53 regulates PAI-1 expression to promote aging-related atrial fibrosis, contributing to the AF progression. This may be the potential mechanism for the increased aging-related fibrosis and susceptibility to AF in the elderly patients.

Materials and Methods

Isolation and culture of human atrial fibroblasts

Human atrial fibroblasts were isolated and cultured as previously described in (Gao et al., 2023). The studies were conformed to the Helsinki Declaration and were approved by the ethics committee of the Guangdong Provincial People’s Hospital, Guangdong Academy of Medical Sciences (No. GDREC20160128H). All subjects were given written informed consent. Patients with pneumonia or other infectious diseases were excluded. Atrial appendages (AAs) were obtained from individuals undergoing cardiopulmonary bypass surgery or thoracoscopic surgery in Guangdong Provincial People’s Hospital (Guangzhou, Guangdong, China). The AAs were fixed in 4% formalin for Masson trichrome staining, or frozen in liquid nitrogen and stored at −80 °C for Western blot analysis or used for cell culture.

The tissues were dissected free of fat, washed with phosphate-buffered solution (PBS), minced into 1–2 mm3 pieces, and evenly inoculated into culture bottles of 25 cm2. After 2 h of incubation, 5 mL of complete cell culture medium (Fibroblast Basal Medium, (Lonza, CC-3131), 10% fetal bovine serum (Gibco; Thermo Fisher Scientific, Waltham, MA, USA), Insulin (Lonza, CC-4021WW), rhFGF (Lonza, CC-4065WW), GA-1000 (Lonza, CC-4081WW)) was delicately added. The tissues continued to be cultured at 37 °C in a humidified incubator of 5% CO2. The atrial fibroblasts crawled from the around tissue fragments and cultured for up to passage14 (P14) to establish the senescence cell model.

Processing of high-throughput sequencing (HTS) data and identification of differentially expressed genes (DEGs)

P4 (young) and P14 (senescent) human atrial fibroblasts were used for mRNA-seq analysis (Guangzhou Epigenome Company, Guangzhou, China). The RNA-seq procedure utilized the VAHTS Stranded mRNA-seq Library Prep Kit for Illumina V2. The sequencing reads were aligned to the GRCh38 reference genome using Hisat2 (Pertea et al., 2016), and gene counts were determined using featureCounts (Liao, Smyth & Shi, 2014). Differential gene expression analysis between P4 and P14 human atrial fibroblasts was visualized using the “pheatmap,” “ggplot2,” and “stats” packages (Turissini et al., 2018).

Processing of microarray data and identification of DEGs

To explore the differences between AF and sinus rhythm (SR), we obtained three microarray datasets (GSE31821, GSE115574, and GSE41177) from the GEO database. These datasets included varying numbers of AF and SR patients, GSE31821 (four AF and two SR patients), GSE115574 (28 AF and 31 SR patients), and GSE41177 (16 AF and three SR patients). The raw data files (*.CEL) were imported using the “affy” package (Gautier et al., 2004), and preprocessing and normalization were performed using the robust multi-array average (RMA) function. Batch effects were addressed using the ComBat function from the “sva” R package (Leek et al., 2012), and data were annotated based on the GPL570 platform. The differentially expressed genes between AF and SR patients were visualized using the “pheatmap,” “ggplot2,” and “stats” packages.

Functional enrichment analysis

To investigate the potential biological functions of SERPINE1, we evaluated its median expression level in the P14 samples from the RNA-seq dataset and categorized the 6 samples (P14) into high and low expression groups. Gene set enrichment analysis (GSEA) was conducted using the ’clusterProfiler’ package (Yu et al., 2012) to identify enriched gene sets associated with SERPINE1 (high vs low). The GSEA results were visualized using the “ggplot2” package. Statistically significant outcomes were determined based on —NES—>1 and FDR <0.25 using gene sets within the C2 section of the msigdb database (Luo et al., 2020).

Investigation of clinical significance and correlation of SERPINE1, EP300, CDKN1A, and TP53, and construction of co-expression networks

To assess the clinical significance of SERPINE1 and its associated genes, Receiver operating characteristic (ROC) analysis was performed using the ‘pROC’ package, with area under the curve (AUC) values above 0.7 indicating good classifier models (Robin et al., 2011). Spearman’s correlations were calculated for each gene, and network visualization was achieved using the ‘ggraph’ package. Protein interaction networks were constructed using the web-based gene network prediction tool GeneMania (Warde-Farley et al., 2010).

Animals

The animal protocol was approved by the Research Ethics Committee of Guangdong Provincial People’s Hospital, Guangdong Academy of Medical Sciences (approval No. GDREC2016128A). C57BL/6 male mice were purchased from the Laboratory Animal Center of Guangzhou University of Chinese Medicine (Guangzhou, China; License: SCXK, Guangdong, 2018-0034). Mice were kept under pathogen-free conditions and kept in a temperature- and humidity-controlled environment on a 12-hour light/dark cycle with free access to food and water until they were 18–20 months old, to establish the aging model. The experimental unit was single mouse. Twenty-nine mice were randomly divided into four groups. Eight mice were performed at the age of 5 months as the young group. The other three groups (seven mice per group) were treated with curcumin skim milk powder pill (50 or 100 mg/kg/day curcumin; Cayman Chemical, Ann Arbor, MI, USA) or equal weight skim milk powder pill as placebo from the age of 12 months until the age of 18-20 months. p300 −/+ mice were generated by crossing p300 floxed mice with CAG-cre mice (Jiangsu GemPharmatech Co., Ltd, Jiangsu, China) (Gao et al., 2023). Floxed, but Cre-negative, littermates were used as experimental controls (wild type, WT). There were four groups (total twenty-four mice), including 7 months or 18 months p300 (−/+) heterozygous mice, and 7 months or 18 months WT mice (six mice per group). Only the feeding personnel know the detailed grouping. Mice were not excluded from this study and confounders were not controlled.

Electrophysiology in vivo

Mice were anesthetized by intraperitoneal injection of 1% sodium pentobarbital (60 mg/kg) and attached to a thermostatic heating plate (RWD Life Science Inc, Shenzhen, China) to maintain body temperature at 37.0 °C. iWorx Data acquisition and analysis system (iWorx Systems, Dover, NH, USA) was used to record and analysis the surface and intracardiac ECG of mice. Limb lead electrodes were inserted under the skin of the mice to record surface ECG. An incision was made in the neck of mouse to expose the left internal jugular vein. Under the guidance of surface and intracardiac ECG, the electrode catheter was inserted in the left atrium through the vein. The pacing threshold was determined by incrementally increasing the amplitude with a pulse width of 1ms and a frequency 10 times/min faster than the basal heart rate until atrial capture occurred. For induction of AF, transvenous rapid atrial pacing techniques was performed. The atrial stimulation parameters were as follows: two-fold pacing threshold, a frequency of 10 Hz, a pulse width of 1 ms, and a duration of 6 s. The pacing was repeated 10 times. AF is defined as an irregular RR rhythm lasting more than 1s and not accompanied by clear P waves on ECG. AF inducibility was defined as the percentage of successful inductions of AF. The duration of AF was defined as the time interval between the onset and the spontaneous termination of AF. At the end of the experiment, mice were euthanized by cervical dislocation and heart tissues were collected.

Isolation and culture of mouse atrial fibroblasts

Primary mouse atrial fibroblasts were isolated from the atrial appendages of C57BL/6 mice (3 weeks old). The atrial appendages were separated from the hearts after mice were euthanized by cervical dislocation. And the atrial appendages were cut into ∼1 mm3 pieces. The pieces were digested with 0.25% EDTA-trypsin (Gibco) using a rotor in a flask. All digestive fluid was collected and centrifuged at 1,000 rpm for 5 min. The pellet was dissolved and seeded into T25 cell culture flasks. The mouse atrial fibroblasts were cultured in Dulbecco’s Modified Eagle’s Medium (DMEM; Gibco, Thermo Fisher Scientific) containing 4.5 g/L D-glucose, 10% fetal bovine serum (Gibco, Thermo Fisher Scientific), penicillin (100 U/ml), and streptomycin (100 µg/ml) in a humidified atmosphere composed of 95% O2 and 5% CO2 at 37 °C.

Curcumin (Cayman Chemical, Ann Arbor, MI, USA) and C646 (Merck, Darmstadt, Germany) were used to inhibit p300, respectively. shRNA plasmid and siRNA specific for human/mouse p300 and p53 (Shanghai Genechem Co. Ltd, Shanghai, China), were transfected into human atrial fibroblasts or mouse atrial fibroblasts using lipofectamine 2000 reagent (Invitrogen) to knock down the gene expression, respectively.

Western blot analysis

Atrial fibroblasts or atrial tissue were homogenized in RIPA lysis buffer (Beyotime Biotechnology, Shanghai, China). After centrifuged at 12,000 rpm for 15 min at 4 °C, lysates were used for protein quantification by Bicinchoninic Acid (BCA) Protein Assay Kit (P0009; Beyotime, Jiangsu, China). The proteins (30 µg) were fractionated on 8% or 10% SDS–polyacrylamide gels and transferred to PVDF membranes (Millipore, Merck, Germany), and blocked with 5% non-fat milk in Tris-buffered saline Tween (TBST) for 1 h at room temperature, then washed 3 times with TBST. After that, membranes were incubated overnight at 4 °C with primary antibodies (Please see details in Supplementary materials). The membranes were then washed with TBST and incubated for 1 h with horseradish peroxidase (HRP)-conjugated secondary antibody. Ultimately, specific protein bands were visualized and detected using electrochemiluminescence (ECL) chemiluminescence system (Merck Millipore, Darmstadt, Germany).

Senescence-associated β-galactosidase staining

At pH 6.0, senescent cell showed significant β-galactosidase activity. In this study, the senescence of atrial fibroblasts and atrial tissue was determined using the Senescence β-Galactosidase Staining Kit (CAT no: #9860; Cell Signaling Technology, Danvers, MA, USA) following the manufacturer’s protocol.

Masson trichrome staining and immunofluorescence

In this study, atrial appendages of C57BL/6 mice were subjected to Masson’s trichrome staining to analyze collagen accumulation in atrial tissue. Masson trichrome staining was performed with the Masson dye solution set (Servicebio, Wuhan, China, G1006) according to the manufacturer’s instruction. The blue color of the tissue indicates the presence of collagen fibers.

Immunofluorescence was performed to observe the distribution of extracellular collagen I in human atrial fibroblasts of different passages. Passage 3 (P3) and passage 11 (P11) human atrial fibroblasts were respectively seeded onto confocal microscope-specialized cover glass and cultured for 2 days allow native collagen to form fibrous structures. After washed with PBS for two times, cells were fixed with 4% paraformaldehyde for 15 min, then rinsed three times with PBS. 4% BSA was used to blocked the cells for 30 min at room temperature. Cells were incubated with Anti-Collagen I antibody (1:50) overnight at 4 °C. After rinsed three times with PBS, cells were incubated with secondary antibodies for 1 h at room temperature protected from the light. After rinsing in PBS, Gold antifade reagent with DAPI (Invitrogen Cat #S36938) was added to counterstain nuclei. The fluorescent images were acquired by laser scanning confocal microscope (SP5-FCS; Leica, Wetzlar, Germany) in a light-protected environment. Image J (ver.1.46) was used for immunofluorescence quantification.

Statistical analysis

Data were presented as means with standard error of mean (SEM). The data were analyzed with SPSS Statistics 20.0. Comparisons between two groups were carried out by unpaired two-tailed Student’s t-test. One-way analysis of variance (ANOVA) was applied for comparison in multi-groups. The LSD and SNK test were used for pairwise comparisons. AF inducibility was compared by Chi-square test. P < 0.05 was considered statistically different.

Results

mRNA-seq and bioinformatics analysis revealed the connections among SERPINE1 (PAI-1), EP300 (p300), CDKN1A(p21), and TP53(p53) in senescent atrial fibroblasts and patients with AF

To commence, we standardized and normalized the raw mRNA-seq data. Subsequently, a heatmap analysis was conducted to juxtapose the gene expression levels of atrial fibroblasts at P4 and P14, as depicted in Fig. 1A. Remarkably, SERPINE1 (p-value 0.03) and EP300 (p-value 0.04) exhibited statistically significant disparities, while TP53 and CDKN1A also exhibited a measure of variation, as exemplified in Figs. 1B–1C. To explore the potential biological function of SERPINE1 in aging individuals, we performed GSEA. The high SERPINE1 expression group revealed enrichment in adherens junction, p53 signaling pathway, and TGF-β signaling pathway, which were intricately linked to the process of aging. Figs. 1D–1G portrays the heatmap correlation analysis between SERPINE1 and EP300, CDKN1A, TP53, TGFBR1 and SMAD2.

Figure 1 The connections among SERPINE1, EP300, CDKN1A, and TP53 in senescent atrial fibroblasts and patients with AF.

(A) Heatmap of standardized and normalized initial RNA-seq data. (B) Heatmap displaying the expression profiles of SERPINE1, CDKN1A, TP53, and EP300 in the P4 (young) and P14 (senescent) human atrial fibroblasts. (C) Violin plots and box plots illustrating the differences in gene expression between the P4 (young) and P14 (senescent) human atrial fibroblasts. (D) GSEA identifying the potential biological functions of SERPINE1. (E–G) Heatmaps showing significantly enriched genes in the regulation of adherens junction, p53 signaling pathway, and TGF-β signaling pathway terms. (H) Heatmap presenting the expression profiles of SERPINE1, CDKN1A, TP53, and EP300 in the SR and AF groups. (I) Violin plots and box plots displaying the differences in gene expression between the SR and AF groups. (J) ROC analysis of individual SERPINE1, EP300, CDKN1A, and TP53 in merged datasets, with the x-axis representing sensitivity and the y-axis representing 1-specificity. The classification performance is represented by the AUC of the ROC. (K) Correlation analysis of the relevant genes conducted in the AF group, using data from 32 AF samples. (L) Co-expression network of SERPINE1, EP300, TP53, and CDKN1A.

Moreover, we conducted an analysis of three independent microarray datasets (GSE31821, GSE115574, and GSE41177) to investigate the role of EP300, CDKN1A, TP53, and SERPINE1 in the context of AF and SR. The analysis divulged an elevated expression of EP300 (p-value 5.5e−03), TP53 (p-value 0.01), CDKN1A (p-value 3.1e−03) and SERPINE1 (p-value 9.4e−03) in AF patients (Figs. 1H–1I). For assessing the discriminative capacity of EP300, CDKN1A, TP53, and SERPINE1 between AF and SR patients, we employed ROC analysis. The analysis yielded AUC values of 0.731, 0.742, 0.711, and 0.714, respectively, signifying their potential as classifiers (Fig. 1J). Importantly, Fig. 1K illuminated a significant correlation between SERPINE1 and the remaining triumvirate of genes, namely EP300, TP53, and CDKN1A. The complete set of four genes was submitted to GeneMANIA to reconstruct their interaction network. The results revealed that SERPINE1, EP300, TP53, and CDKN1A might be mediated by TGFB induced factor homeobox 1 (TGIF1), as shown in Fig. 1L.

p300 regulated PAI-1 protein expression in senescent human atrial fibroblasts

Primary human atrial fibroblasts were cultured to P11 to construct a cellular replicative senescence model. SA-β-gal staining was performed to observe the ratio of senescence cells in P3 and P11 cells. As shown in Fig. 2A, senescent cells were flatter and enlarged, and more perinuclear blue, compared with young cells (P3), the ratio of senescent cells in P11 human atrial fibroblasts increased. Moreover, greater amounts of collagen I were secreted into the extracellular matrix, consistent with both increased production and polymerization of collagen I fibers in P11 cells compared to P3 cells (Fig. 2B). This indicated that increased fibrosis related protein was accompanied with the senescence of atrial fibroblasts.

Figure 2 Increased p53/p21 and PAI-1 were regulated by p300 in senescent human atrial fibroblasts.

(A) Representative SA-β-gal staining and positive cells analysis of young (P3) and senescent (P11) human atrial fibroblasts. Scale bars, 100 µm. (B) Representative immunofluorescence staining and fluorescence intensity of Col1A1 in young and senescent human atrial fibroblasts. Scale bars, 50 µm. (C–D) Representative immunoblots and densitometric analysis of p300, p53, p21 and PAI-1 in young (P3) and senescent human atrial fibroblasts (P11) treated with curcumin (6, 9, or 12 µM) or C646 (5, 10, 15, or 25 µM). ∗P < 0.05, ∗∗P < 0.01; Data are mean ± SEM.

In addition, the protein expression levels of p300, p53/p21 and PAI-1 were also increased in senescent (P11) human atrial fibroblasts (Fig. 2C). Furthermore, curcumin, a natural specific inhibitor of p300/CBP protein expression and histone acetyltransferase (HAT) activity, was used to investigate the role of p300 in the regulation of p53/PAI-1 and cell senescence. The expression of p300 decreased, accompanied by a gradual decrease in the levels of p53/p21 and PAI-1 after treated with curcumin at the concentration of 6, 9, and 12 µmol/L (Fig. 2C). Similar results were obtained when P11 human atrial fibroblasts were treated with selective p300 inhibitor C646. The expression levels of p300, p53 and PAI-1 decreased with the increase of C646 concentration (Fig. 2D). These results suggest that the p53 and PAI-1 was regulated by p300 during cellular senescence, p300 may not only contribute to cellular senescence but also participate in the regulation of fibrosis related protein.

p300 or p53 knockdown reversed cellular senescence and decreased PAI-1 expression

To further determine the role of p300 in the regulation of PAI-1 during senescent human atrial fibroblasts, p300shRNA was used to knockdown p300 protein levels in senescent human atrial fibroblasts (P11). As shown in Figs. 3A–3B, p300 knockdown significantly decreased the ratio of SA-β-gal positive cells and the protein expression of p53/p21 and PAI-1 in P11 cells.

Furthermore, to explore the specific molecular mechanism of p53 in the regulation of PAI-1, RNA interference (RNAi) technology was also used to knock down the level of p53 in P11 human atrial fibroblasts. The Western blot results are shown in Fig. 3C. The expression of p21 and PAI-1 decreased significantly after p53siRNA transfection of P11 human atrial fibroblasts. These results indicated that PAI-1 was regulated by p300/p53 in senescent human atrial fibroblasts.

p300 also mediated mouse atrial fibroblasts senescence and fibrosis through p53/PAI-1 pathway

We also determined the role of p300 in the regulation of p53/PAI-1 pathway in senescence and fibrosis in mouse atrial fibroblasts. Similarly, SA-β-gal staining confirmed that the ratio of senescent P11 mouse atrial fibroblasts was significantly increased when compared to P3 control cells (Fig. 4A). The expression of p300, p53 and PAI-1 were also increased in senescent mouse atrial fibroblasts (Fig. 4B). Inhibition of p300 using curcumin or p300 siRNA could reduce the ratio of senescent cells and the expression of p53 and PAI-1 in P11 cells (Figs. 4B–4D). Subsequent, p53 knockdown with siRNA also reduced the protein expression of p21 and PAI-1 (Fig. 4E). These results revealed that p300/p53 also mediated the PAI-1 expression in senescent mouse atrial fibroblasts.

The susceptibility to AF was increased in aged mice with the activation of p300/p53/PAI-1 pathway

Further studies were performed in aged mouse model. The electrophysiological examination data showed that PR interval was significantly prolonged in the aged group (18-month mice) compared with the young group (5-month mice). SNRT and CSNRT were also significantly increased in the aged mice compared with the young group (Table 1), which indicated delayed atrioventricular conduction and abnormal sinus node function. Transvenous rapid atrial pacing experiment results showed that AF inducibility increased significantly with mice aging (Fig. 5A).

Figure 3 P300/p53 mediated the regulation of PAI-1 expression in senescent human atrial fibroblasts.

(A) Representative SA-β-gal staining and positive cells analysis of senescent human atrial fibroblasts (P11) treated with or without p300shRNA. Scale bars, 100 µm. (B). Representative immunoblots and densitometric analysis of p300/p53/p21 and PAI-1 in senescent human atrial fibroblasts (P11) treated with p300 shRNA. (C) Representative immunoblots and densitometric analysis of p53/p21 and PAI-1 in senescent human atrial fibroblasts (P11) treated with p53 siRNA. ∗P < 0.05, ∗∗P < 0.01; data are mean ± SEM.

Figure 4 P300/p53 also regulated PAI expression in senescent mouse atrial fibroblasts.

(A) Representative SA-β-gal staining and positive cells analysis of young (P3) and senescent (P11) mouse atrial fibroblasts. Scale bars, 100 µm. (B). Representative immunoblots and densitometric analysis of p300, p53 and PAI-1 in young (P3) and senescent mouse atrial fibroblasts (P11) treated with curcumin (6, 9, or 12 µM). (C) Representative SA- β-gal staining and positive cells analysis of senescent mouse atrial fibroblasts (P11) treated with or without p300siRNA. Scale bars, 100 µm. (D–E) Representative immunoblots and densitometric analysis of p300/p53/p21 and PAI-1 in senescent mouse atrial fibroblasts (P11) treated with p300 siRNA or p53 siRNA. ∗P < 0.05, ∗∗P < 0.01; data are mean ± SEM.

Table 1 General characteristics and electrophysiological analysis of mice.

	5 m	18 m	18 m + Cur (50 mg/kg/d)	18 m + Cur (100 mg/kg/d)	
N	8	7	7	7	
PWD (ms)	18.5 ± 0.98	19.29 ± 1.57	21.57 ± 1.45	19.00 ± 1.07	
PR interval (ms)	41.50 ± 1.94	50.57 ± 1.91**	45.14 ± 1.20#	44.71 ± 1.66#	
QRS duration (ms)	12.25 ± 0.65	15.29 ± 1.13	15.14 ± 0.59	14.29 ± 1.21	
QT interval (ms)	28.50 ± 0.89	29.71 ± 1.17	30.14 ± 0.91	29.14 ± 1.58	
SNRT (ms)	131.13 ± 2.80	225.86 ± 19.17*	195.29 ± 8.28**	178.14 ± 10.95*	
CSNRT (ms)	24.08 ± 1.69	74.77 ± 11.17*	50.51 ± 7.38	34.79 ± 3.26	
AF inducibility (%)	12.5%	71.4%*	14.3%#	14.3%#	
Total AF duration (s)	1.16	246.00	4.92	3.08	
Notes.

Data are mean ± SEM.

PWD P wave duration

SNRT Sinus node recovery time

CSNRT Corrected SNRT

AF Atrial fibrillation

* P < 0.05 vs. 5 m.

** P < 0.01 vs. 5 m.

# P < 0.05 vs. 18 m.

Aggravated atrial fibrosis and higher ratio of aging atrial tissue in 18-month mice compared with 5-month mice, which were detected by Masson staining or SA-β-gal staining (Figs. 5B–5C). As shown in Fig. 5D, compared to 5-month mice, 18-month mice showed an increase in the expression of p300, p53/p21 and PAI-1. Mice were administered oral curcumin (50/100 mg/kg/d) for 6 months until 18 months old. Both 50 and 100 mg/kg curcumin significantly reduced the AF inducibility (Fig. 5A). Moreover, atrial fibrosis and atrial tissue aging degree were improved by 100 mg/kg curcumin (Figs. 5B–5C). The protein expression of p300, p53/p21 and PAI-1 was also reduced in curcumin treated 18-month mice and p300 (-/+) Het 18-month mice (Figs. 5E–5F). The above findings show that p300 inhibitor curcumin can inhibit the activation of p300/p53/PAI-1 signaling pathway, thereby alleviating the atrial aging and fibrosis of aged mice, ultimately improving the age-related AF.

Discussion

Aging is an independent risk factor for AF, but the molecular mechanism of AF caused by aging is unclear. In the present study, mRNA-seq showed a significant increase in p300 and PAI-1 gene expression in senescent human atrial fibroblasts. Then, we confirmed the increased protein expression of p300, p53, p21 and PAI-1 in senescent (P11) human and mouse atrial fibroblasts. Our previous study identified the critical regulatory role of p300/p53 in elderly AF (Gao et al., 2023). Studies have reported that PAI-1 is also a downstream target of p53 and is closely related to aging and fibrosis (Kortlever, Higgins & Bernards, 2006). This suggests that PAI-1 may also be regulated by p300/p53 to participate in age related atrial fibrosis, thus lead to the occurrence and development of AF in the aged. In this study, we found that inhibitor of p300 (Curcumin or C646) and p300 RNAi decreased p53 and PAI-1 expression in senescent human and mouse atrial fibroblasts. Furthermore, intervention of p53 could inhibit the expression of PAI-1 in P11 human and mouse atrial fibroblasts, thereby alleviating cell senescence and fibrosis. Treatment with p300 inhibitor curcumin also reduced the expression of p53/p21 and PAI, thereby reversing age and age-related atrial fibrosis in aged mice, and reducing the susceptibility to AF.

Figure 5 The activation of p300/p53/PAI-1 pathway contributed to the increased AF susceptibility.

(A) Typical surface electrocardiogram (ECG) showing that AF was successfully induced by atrial burst stimulation in mice. And the inducible rate of AF in young and aged mice treated with different dosage of curcumin. n = 7–8. ∗P < 0.05 vs. 5 m mice, #P < 0.05 vs. 18 m mice. (B) Representative Masson staining of atrial specimens obtained from young (5 m) and aged (18 m) mice treated with curcumin (100 mg/kg/d). Scale bars, 50 µm. (C) Representative SA-β-gal staining of atrial tissue of young (5 m) and aged (18 m) mice treated with or without 100 mg/kg/d curcumin. Scale bars, 100 µm. (D) Representative immunoblots and densitometric analysis of p300, p53/ p21 and PAI-1 in in atrial tissues from young (5 m) and aged (18 m) mice (n = 6). (E–F) Representative immunoblots and densitometric analysis of p300, p53/p21 and PAI-1 in in atrial tissues from aged mice treated with different doses of curcumin (50 or 100 mg/kg/d) (n = 6) and from young (7 m) and aged (18 m) p300 (+/+) (wt) and p300 (−/+) (het) mice (n = 6). Data are mean ± SEM. ∗P < 0.05, ∗∗P < 0.01.

Atrial fibrosis is one of the main pathological mechanisms of AF, and aging is an important risk factor for atrial fibrosis. Aging related atrial fibrosis plays an important role in the occurrence and development of elderly AF. Research have indicated that co-transcriptional activator and acetyltransferase p300, as the opposite regulator of histone deacetylase and a driver of senescence via inducing superenhancers, is involved in aging or senescence (Ghosh, 2020; Sen et al., 2019). In addition, recent studies implicate the importance of p300 in extracellular matrix (ECM) remodeling and fibrosis (Ghosh et al., 2013; Ghosh et al., 2000; Ghosh & Varga, 2007). In the cardiovascular system, p300 may promote myocardial fibrosis in the process of pathological hypertrophy of diabetes or hypertension animal models by mediating the acetylation of Smad. Suppression of p300 acetyltransferase activity with natural molecules (curcumin) or synthetic small molecule inhibitors (L002 or C646) leads to improvement of cardiac fibrosis induced by diabetes or hypertension(Bugyei-Twum et al., 2014; Rai et al., 2017; Rai et al., 2019). Our previous research has shown that the aging degree and fibrosis of atrial tissue in elderly patients with AF and aged mice were significantly increased, with a significant increase in the expression of aging related signal pathway proteins p53/p21, accompanied by an increase in the expression of acetyltransferase p300 and fibrosis factors. It was confirmed in the senescent human atrial fibroblasts that p300 participated in aging related atrial fibrosis by activating the p53/Smad3 signaling pathway (Gao et al., 2023). In this study, by using bioinformatics analysis and mRNA-seq sequencing of young and senescent human atrial fibroblasts, we found that PAI-1 may also play an important role in aging induced atrial fibrosis.

PAI-1 is a member of the serpin (serine protease inhibitor) family responsible for suppressing the fibrinolytic activity (Eren et al., 2014). In addition, PAI-1 as a key enzyme regulating extra-cellular matrix (ECM) metabolism which forms the basis on the research field of organ fibrosis, is an important profibrotic factor (Li et al., 2021). PAI-1 inhibits the activation of tPA and uPA to form plasmin, leading to the reduction of ECM degradation and a large amount of ECM deposition which finally promote fibrosis (Rabieian et al., 2018; Ghosh & Vaughan, 2012) including cardiac fibrosis in cardiac diseases (Li et al., 2021; Sillen & Declerck, 2020). Moreover, PAI-1 is a marker of replicative senescence and its level increases with age (Vaughan et al., 2017). Studies have also demonstrated that PAI-1 is a necessary and sufficient downstream of p53-induced replicative senescence, considered an important mediator of cellular senescence. Inhibition of p53 target gene encoding PAI-1 by RNAi technology leads to escape from replicative senescence both in mouse embryonic fibroblasts and human BJ fibroblasts (Kortlever, Higgins & Bernards, 2006). This indicated that PAI-1 is the critical factor contributing to senescence and fibrosis. Taken together, our results suggest that p300/p53 can also promote PAI-1 expression, therefore lead to atrial fibrosis in senescent atrial fibroblasts. This study provides another new and powerful evidence chain for p300/p53 regulating age-related fibrosis.

In addition, hypercoagulable state or prethrombotic state in AF patients cannot be overlooked (Wolf et al., 1978; Lip, 1995). AF leads to thrombosis, which increases the risk of stroke and systemic embolism by three to five times, and this is the main cause of mortality in elderly patients with AF (Escudero-Martínez, Morales-Caba & Segura, 2023). The prevention of thromboembolic complications is an essential part of the treatment of AF (Hindricks et al., 2021; Chiang et al., 2017). It has been demonstrated that decreased fibrinolytic activity is an important factor in the thrombosis of AF (Marín, Roldán & Lip, 2003). PAI-1 is considered to be the key gene in age-related prothrombotic state (Yamamoto et al., 2005), we suggested that the activation of p300/p53/PAI-1 pathway might also be the potential mechanism for the hypercoagulable state of blood in aging-related AF. Anticoagulation is an effective therapy to prevent thromboembolism in patients with AF (Lip et al., 2017). Although great progress has been made in the development of new anticoagulants such as non-vitamin K antagonists and left atrial appendage occlusion surgery (Katsanos et al., 2020), the treatment of AF especially in the elderly is far from ideal. The current drugs do not interdict underlying disease mechanisms (Ang et al., 2020). In this study, we revealed the pivotal role of PAI-1 in aging-related AF and hypercoagulable state. These data provide compelling evidence that suppression of p300 as an upstream therapy to prevent the progression of AF by reducing PAI-1 levels to inhibit the substrates for AF development and thrombosis is considerable for the elderly.

Conclusions

Taken together, we conclude that PAI-1 might play an important role in the pathogenesis and prognosis of age-related AF by regulating aging-related fibrosis and fibrinolytic system, and it is regulated by p300. Upstream therapies to inhibit p300, such as curcumin treatment or p300 knockdown, can improve atrial fibrosis to reduce the AF susceptibility of elderly patients, and improve the hypercoagulable state of blood to reduce the risk of thrombosis. Intervention of p300 is a very feasible new insight for the prevention and treatment of AF in the elderly.

Supplemental Information

Supplemental Information 1 The raw data of RNA-seq

Click here for additional data file.

Supplemental Information 2 The original file used for GSEA analysis and the analysis results for Figs. 1D–1G

Click here for additional data file.

Supplemental Information 3 The details of differentially expressed genes (DEG)

Click here for additional data file.

Supplemental Information 4 The pre-normalized data vs post-normalized data (A, C) were plotted

Heat maps of the differentially expressed genes are in (B) and (D).

Click here for additional data file.

Supplemental Information 5 Supplemental Methods

Click here for additional data file.

Supplemental Information 6 Full (21-point) ARRIVE 2.0 checklist

Click here for additional data file.

Supplemental Information 7 Raw Data for Fig. 2–Fig. 5 and Table 1

Click here for additional data file.

Supplemental Information 8 Full-length uncropped gels/blots (Fig. 2–Fig. 5)

Click here for additional data file.

We would like to thank Prof. Nanette H. Bishopric and Dr. Jianqin Wei for providing p300 flox/flox mice.

Additional Information and Declarations

Competing Interests

Author Contributions

Human Ethics

Animal Ethics

Microarray Data Deposition

Data Availability

The authors declare there are no competing interests.

Yingyu Lai performed the experiments, analyzed the data, prepared figures and/or tables, authored or reviewed drafts of the article, and approved the final draft.

Jintao He performed the experiments, analyzed the data, prepared figures and/or tables, authored or reviewed drafts of the article, and approved the final draft.

Xiaoyan Gao performed the experiments, authored or reviewed drafts of the article, and approved the final draft.

Dewei Peng performed the experiments, authored or reviewed drafts of the article, and approved the final draft.

Huishan Zhou performed the experiments, authored or reviewed drafts of the article, and approved the final draft.

Yuwen Xu performed the experiments, authored or reviewed drafts of the article, and approved the final draft.

Xueshan Luo performed the experiments, authored or reviewed drafts of the article, and approved the final draft.

Hui Yang conceived and designed the experiments, performed the experiments, prepared figures and/or tables, and approved the final draft.

Mengzhen Zhang performed the experiments, prepared figures and/or tables, and approved the final draft.

Chunyu Deng conceived and designed the experiments, prepared figures and/or tables, and approved the final draft.

Shulin Wu conceived and designed the experiments, prepared figures and/or tables, authored or reviewed drafts of the article, and approved the final draft.

Yumei Xue conceived and designed the experiments, prepared figures and/or tables, authored or reviewed drafts of the article, and approved the final draft.

Feng Zhou conceived and designed the experiments, prepared figures and/or tables, authored or reviewed drafts of the article, and approved the final draft.

Fang Rao conceived and designed the experiments, prepared figures and/or tables, authored or reviewed drafts of the article, and approved the final draft.

The following information was supplied relating to ethical approvals (i.e., approving body and any reference numbers):

The studies were approved by the ethics committee of the Guangdong Provincial People’s Hospital, Guangdong Academy of Medical Sciences (No. GDREC20160128H)

The following information was supplied relating to ethical approvals (i.e., approving body and any reference numbers):

The animal protocol was approved by the Research Ethics Committee of Guangdong Provincial People’s Hospital, Guangdong Academy of Medical Sciences (approval No. GDREC2016128A).

The following information was supplied regarding the deposition of microarray data:

The gene expression data is available at NCBI GEO: GSE31821, GSE115574, and GSE41177.

The following information was supplied regarding data availability:

The raw data is available in the Supplemental Files.

The code is available at GitHub and Zenodo:

– https://github.com/Jorth-Jorth/Perrj-code.git.

– Jorth-Jorth. (2023). Jorth-Jorth/Perrj-code: perrj (perrj). Zenodo. https://doi.org/10.5281/zenodo.10049747.

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
