# Peer review of "Involvement of plasminogen activator inhibitor-1 in p300/p53-mediated age-related atrial fibrosis"

_PeerJ, doi:10.7717/peerj.16545_

## Round 0.1 · original submission · Minor Revisions

It is my opinion as the Academic Editor for your article - Involvement of plasminogen activator inhibitor-1 in p300/ p53-mediated age-related atrial fibrosis - that it requires a few Minor suggestions from Reviewers. Potential and nice work done by authors.

Reviewer 1 ·

Basic reporting

General Assessment:
The manuscript presents a comprehensive study focusing on the role of SERPINE1, EP300, CDKN1A, and TP53 genes in aging atrial fibroblasts and their potential link to atrial fibrillation (AF). The study uses a robust methodology and offers some interesting insights. However, there are several areas that require further clarification, elaboration, or revision.

Major Comments:

- Statistical Methods: It is essential to describe in more detail the statistical methods used in the study. Did you apply any corrections for multiple comparisons? What statistical software was used?

- Mechanistic Insights: The manuscript would be strengthened by a more detailed discussion of the underlying mechanisms connecting these genes to aging and AF.

- Inhibitors Used: The study employs curcumin as an inhibitor for p300, and mentions another inhibitor, C646. What are the specificities of these inhibitors, and could they have off-target effects?


Summary:
Overall, the study is promising and has the potential to contribute valuable information to the field. However, the manuscript needs major revisions for clarity, depth, and completeness of the scientific content presented.

I recommend resubmission after major revisions.

Experimental design

No comment

Validity of the findings

No comment

Additional comments

No comment

·

Basic reporting

no comment

Experimental design

no comment

Validity of the findings

no comment

Additional comments

no comment

---

## Round 0.2 · accepted · Accept

Authors did all corrections/updates based on the reviewers' comments. Both reviewers are satisfied with the current updated version.

Reviewer 1 ·

Basic reporting

The authors have addressed all my concerns.

Experimental design

No comments.

Validity of the findings

No comments.

Additional comments

No comments.

·

Basic reporting

N/A

Experimental design

N/A

Validity of the findings

N/A

Additional comments

N/A